# Can VLMs Diagnose and Recover from VLA Manipulation Faults?

Bowen Yan [* 1]  Jiahao Xiao [* 1]  Kehui Liu [1]  Jianbo Zhang [2]  Zicheng Zhang [1]  Qi Jia [1]  Zhongjie Jia [2]  Haoming Song [2]  Chunyi Li [3]  Bin Zhao [1]  Guangtao Zhai [1]

## Abstract

Existing VLA models frequently fail in robotic manipulation tasks, with poorly structured fault types that often require expert diagnosis. While VLMs offer strong explanatory capabilities, their effectiveness in assisting VLAs is limited by their unclear role in diagnostics and inadequate collaboration mechanisms. To address this, we introduce VLA-FixBench, a fault evaluation dataset that spans perception, planning, and control failures, and provides annotations for task stages, fault types, and spatiotemporal repair strategies. We further propose FaultEval, a static-to-dynamic-to-real evaluation framework that benchmarks 20 VLMs across multiple fault-related dimensions. Building on these insights, we design a VLM–VLA collaboration mechanism that localizes spatiotemporal deviations and rolls back task execution to enable targeted recovery. Experiments show that FaultEval reliably characterizes VLM-based closed-loop diagnosis and repair. The upper-bound analysis using human expert intervention shows that an idealized feedback loop can improve task success rates by 13% on LIBERO and 35% on real-world robots. Our code, benchmark, and project page will be publicly released at: https://kakigo.github.io/VLA-FixBench/

## 1. Introduction

With the rapid advancement of embodied intelligence, Vision-Language-Action (VLA) models have demonstrated increasing advantages in scenarios such as industrial assembly, logistics sorting, and household services. However, due to current technical limitations, existing VLA models frequently experience failures in robotic manipulation tasks. Fault diagnosis heavily relies on expert knowledge and manual intervention, making it difficult to scale. The diagnostic process is often subjective, lacks standardized evaluation criteria, and suffers from insufficient responsiveness for real-time systems. These challenges underscore the need for principled, standardized benchmarks and automated recovery mechanisms that can evaluate and improve failure handling in a reproducible and real-time manner.

Vision–Language Models (VLM) exhibit semantic understanding capabilities in task explanation and strategy guidance (Wang et al., 2025b; Zhang et al., 2025b;c), raising a natural question: Can VLMs effectively assist VLAs in fault diagnosis and correction? Prior evaluation efforts have primarily focused on static task understanding, such as spatial reasoning and visual question answering (Zhang et al., 2025d; Chen et al., 2025), with limited study of multi-level VLM-VLA collaboration (Yang et al., 2025b). Although existing benchmarks capture certain VLM capabilities (Wang et al., 2025a; Wu et al., 2025c;b; Zhang et al., 2025a), their impact on improving VLA performance remains unclear (Grislain et al., 2025). Moreover, current VLM-VLA interactions are largely instruction-based and lack a unified closed-loop framework for diagnosis and recovery (Yang et al., 2025a; Thoduka et al., 2024). As a result, existing methods face key limitations: insufficient focus on VLM-VLA collaboration with no standardized interfaces (Yang et al., 2025b; Chen et al., 2024), coarse datasets lacking fine-grained fault and repair annotations, the absence of a unified evaluation framework for VLM-assisted repair in embodied manipulation, and inconsistent fault taxonomies that fail to decouple perception, control, and planning errors. To address these challenges, we introduce **VLA-FixBench**, a benchmark for VLM-assisted VLA fault diagnosis and recovery, with over 6,000 annotated failure cases across perception, planning, and control. To our knowledge, VLA-FixBench is the first benchmark that jointly evaluates static diagnosis, temporal rollback localization, geometric correction, simulation recovery, and real-robot recovery for VLM-assisted VLA manipulation faults. Based on VLA-FixBench, we propose **FaultEval**, a unified static-to-dynamic-to-real evaluation framework that assesses VLM performance in fault identification, severity estimation, temporal localization, spatial correction, and

---

*Equal contribution [1]Shanghai AI Laboratory, Shanghai, China [2]Shanghai Jiao Tong University, Shanghai, China [3]Nanyang Technological University, Singapore. Correspondence to: Chunyi Li <lichunyi@pjlab.org.cn>, Guangtao Zhai <zhaiguangtao@pjlab.org.cn>.

*Proceedings of the $43^{rd}$ International Conference on Machine Learning*, Seoul, South Korea. PMLR 306, 2026. Copyright 2026 by the author(s).

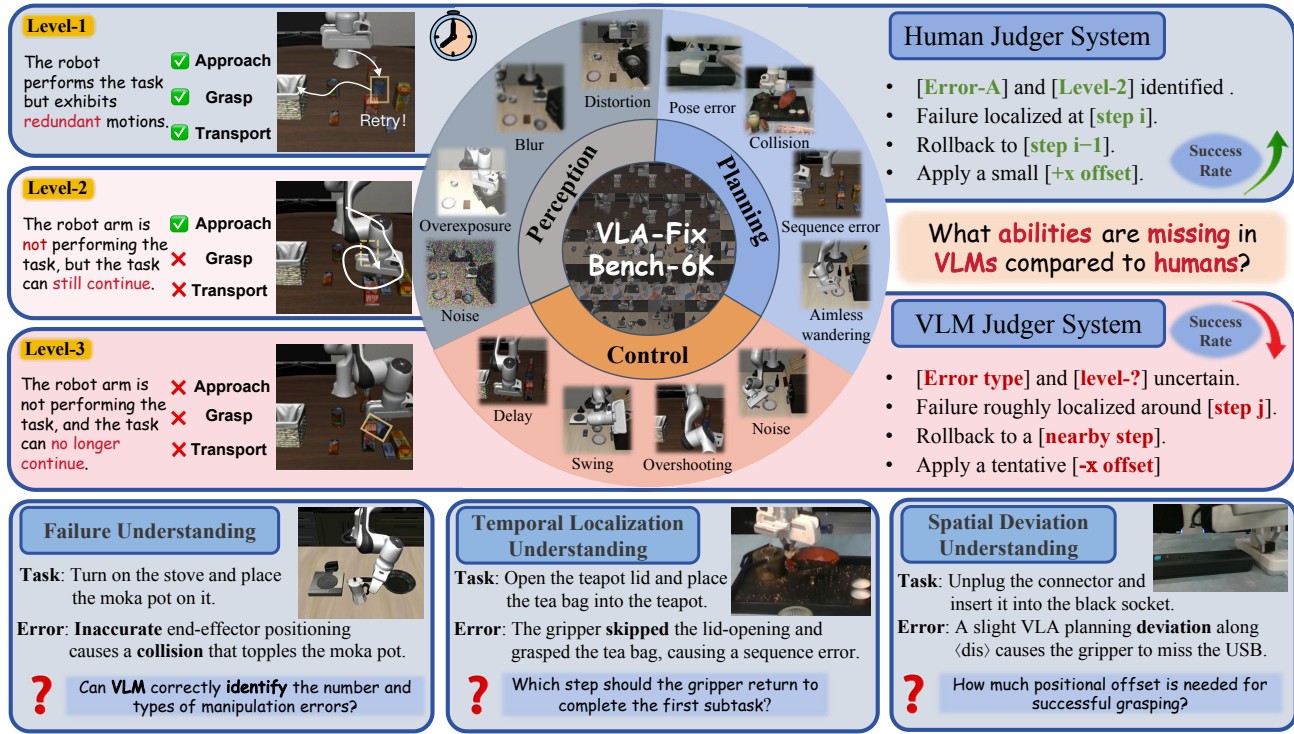

*Figure 1.* **Overview of VLA-FixBench**, Center: Hierarchical failure types in **Perception**, **Planning**, and **Control**. Left: Severity definitions ranging from redundant motions (Level-1) to catastrophic failures (Level-3). Right: Performance disparity between Human and VLM judgers in closed-loop error recovery. Bottom: Three evaluation dimensions: Failure identification, Temporal localization for rollback, and Spatial deviation estimation for corrective action.

closed-loop recovery. We further construct a VLM–VLA collaboration mechanism that enables fault detection, rollback, and action repair during VLA execution. Based on VLA-FixBench, we additionally construct and validate a **VLM–VLA collaboration mechanism** that enables real-time fault detection, rollback, and action repair during VLA execution. Although the automated VLM–VLA loop still provides limited gains due to the reasoning-execution gap, human-expert upper-bound intervention demonstrates the potential of the proposed detect–rollback–correct mechanism. By providing accurate recovery parameters, expert intervention improves task success rates on LIBERO (Liu et al., 2023)and real-world robots. These results suggest that rollback-based recovery can substantially improve task robustness when fault localization and corrective actions are sufficiently accurate.

Our main contributions are summarized as follows:

- **VLA-FixBench**, a robotic manipulation fault evaluation dataset comprising 6k failure cases covering perception, control, and cognition errors, with fine-grained annotations of sub-task stages, error types, and repair strategies.

- **FaultEval**, a unified evaluation framework that benchmarks VLMs across fault recognition, localization,

causal reasoning, repair suggestion, and temporal consistency, and evaluates 20 representative vision language models.

- **VLM-VLA collaboration mechanism**, a VLM–VLA collaboration mechanism that leverages VLM-based fault detection and repair guidance to form a closed-loop perception, decision and execution feedback system, achieving an average 13% improvement in task success rates across multi-task evaluations.

## 2. Related Works

### 2.1. Robotic Failure Diagnosis and Recovery

Early studies on robotic fault diagnosis primarily focus on low-level anomaly detection (Zhao et al., 2025), such as joint torque deviation, force–torque inconsistency, or trajectory tracking errors (Thoduka et al., 2024).These approaches are effective for detecting deviations but lack task-level interpretability and offer no actionable repair guidance. To bridge low-level signals and task execution, some works analyze failures in specific manipulation tasks. As a result, existing failure analyses remain fragmented and lack a unified framework for systematic evaluation across tasks and models (Lin et al., 2025).Classical learning-based ap-

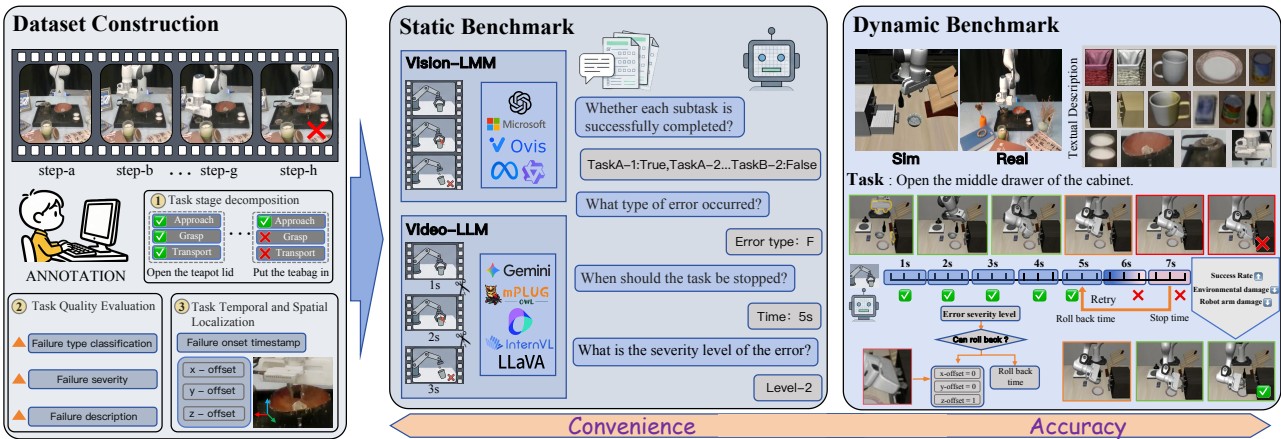

*Figure 2.* **Overview of the VLA-FixBench framework**. The pipeline includes: (Left) Dataset Construction with hierarchical task decomposition and spatio-temporal failure annotations; (Middle) Static Benchmark for evaluating VLMs' diagnostic reasoning; and (Right) Dynamic Benchmark for assessing closed-loop performance and error recovery. The bottom axis indicates the trade-off between evaluation convenience and physical accuracy.

proaches employ supervised classifiers or temporal models to predict failure states from sensory inputs. While effective within constrained settings, these models struggle to generalize to heterogeneous failure types and offer limited interpretability or corrective guidance. Reinforcement learning has been explored for failure recovery by learning corrective behaviors through interaction, including recent efforts that guide agents back to in-distribution states after Out-Of-Distribution (OOD) failures (Kim et al., 2025). While effective in specific domains (Li et al., 2026), such methods typically rely on task rewards or policy-level supervision, embedding recovery implicitly in learned behaviors. Consequently, they offer limited interpretability, generalize poorly across diverse manipulation tasks, and lack explicit task-level fault diagnosis and repair representations. Recent VLMs demonstrate strong capabilities in visual understanding, explanation, and high-level reasoning, enabling applications in task planning and instruction following. Despite promising results, the role of VLMs in assisting VLAs during failure remains unclear (Dai et al., 2024; Pacaud et al., 2025).As a result, it is difficult to assess whether VLMs genuinely improve robotic robustness or merely provide plausible post-hoc explanations.

### 2.2. Benchmark and Failure Evaluation of VLM

Recent advancements in Vision-Language-Action (VLA) models have significantly enhanced the capabilities of embodied agents in diverse manipulation tasks (Team, 2025; Duan et al., 2024). The fundamental principle of VLA models lies in cross-modal grounding, where visual perceptions are interleaved with textual prompts to condition the robot policy. Architectures like OpenVLA (Kim et al., 2024) and GR00T N1 (NVIDIA et al., 2025) utilize frozen vision-language backbones augmented with lightweight ac-

tion heads to maintain broad semantic knowledge while fine-tuning on robotic trajectories. These models represent actions as a structured sequence within a generative framework. However, as task complexity increases, these models frequently suffer from error accumulation in multi-stage execution, where minor initial deviations often escalate into irreversible system failures (Khanna et al., 2025). While existing benchmarks like LIBERO (Liu et al., 2023)provide rigorous environments to assess task success rates , they largely overlook the underlying failure behaviors. Current evaluation standards remain predominantly binary, lacking the fine-grained diagnostic granularity required to categorize specific faults or understand the mechanisms of error recovery. This gap necessitates a more comprehensive framework to analyze and rectify VLA model vulnerabilities. In contrast to prior work, we focus on task-level, interpretable, and recoverable failures in robotic manipulation. We introduce a unified benchmark and evaluation framework that systematically characterizes failure types, severity, and spatiotemporal repair behaviors, and explicitly measures how VLMs contribute to closed-loop fault recovery in VLAs.

## 3. Construction of VLA-FixBench

**Hierarchical Failure Classification** To diagnose VLA failures, we design a two-level hierarchical classification (Figure 1). The first level categorizes failures by functional stage—perception, control, and planning—to localize errors. The second level evaluates failure severity from the perspective of recoverability, considering task continuity, motion quality, and environmental impact. Failures are grouped into four levels: no noticeable failure, minor disturbance, self-recoverable failure, and unrecoverable failure requiring human intervention. This scheme unifies functional attribution with severity assessment, providing a consistent

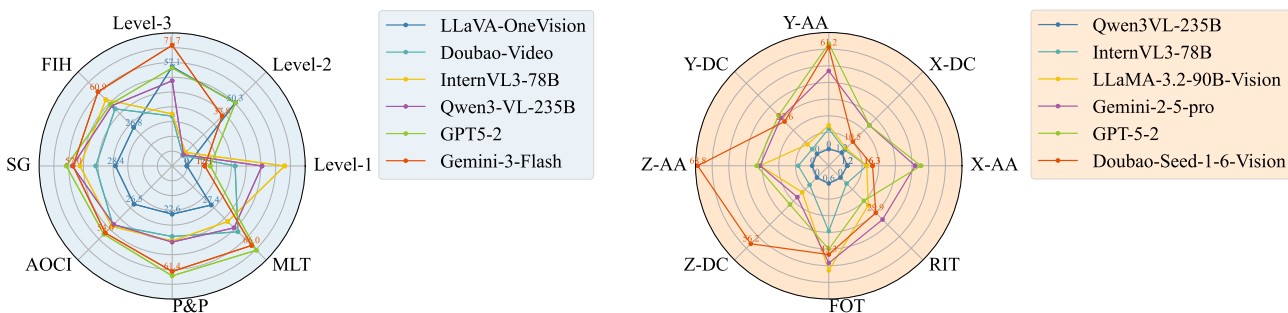

*Figure 3.* Left: Static diagnostic accuracy across different task domains and severity levels. Right: Dynamic diagnostic performance, including Geometric Correction Accuracy (GCA) and Temporal Localization Accuracy (TLA).

framework for VLA failure analysis.

**Multi-dimensional Annotation** We develop a fine-grained annotation framework to construct a high-resolution failure map across three integrated dimensions: temporal, spatial, and semantic. Specifically, the framework incorporates: (i) Sub-task Success, evaluating stage-wise completion; (ii) Qualitative Evaluation, identifying fault types and severity levels; (iii) Temporal Localization, determining failure onset timestamps; and (iv) Spatial Characterization, measuring the deviations of end-effectors or key objects.

**Dataset Statistics.** VLA-FixBench comprises 6,034 task execution episodes collected from 40 simulated and two real-robot environments. The dataset systematically captures a wide spectrum of failure types, stages, and severity levels. Frequent failures such as *Jitter*, *Grasp Instability*, and *Stagnation* appear in over 25% of samples, while rare failures (e.g., *Wrong Placement*, *Target Occluded*) occur in less than 7%, supporting edge-case analysis. The detailed statistics are provided in Appendix A.

## 4. FaultEval Evaluation Framework

We propose a dual-track framework combining a static "written examination" and a dynamic "road test." The static track evaluates fault identification, severity estimation, and spatiotemporal localization, while the dynamic track assesses closed-loop repair effectiveness and final task success. Together, these complementary tracks balance efficiency and accuracy, enabling systematic model comparison and diagnosis for VLA systems.

### 4.1. Static Evaluation

We formalize the static evaluation as a multi-label prediction task conditioned on an input video $V$ and a linguistic instruction $I$. For each sample $i$, the model generates a diagnostic tuple $\hat{y}_i = \langle f_i, l_i, \mathbf{s}_i \rangle$ comprising the **Fault Type Classification (FTC)** $f_i \in \mathcal{F}$ identifying the specific category of the detected failure, the **Fault Severity Rating (FSR)** $l_i \in \{0, 1, 2, 3\}$ gauging the intensity of

the failure, and the **Subtask Success Evaluation (SSE)** $\mathbf{s}_i = \{s_{i,1}, \ldots, s_{i,m}\}$ representing the binary completion status $s_{i,j} \in \{0, 1\}$ across $m$ sequential subtasks.

To reduce the computational cost and localization difficulty of long-horizon task evaluation, we adopt a stage-wise reconstruction strategy. Each task is temporally segmented into consecutive stages, with visual observations sampled at fixed intervals and provided sequentially to the model. This design enables image-only models to be evaluated under a unified protocol. The detailed breakdown of dimensions is provided in Appendix E.

### 4.2. Dynamic Evaluation

We formalize the dynamic evaluation as a closed-loop recovery task requiring precise temporal and spatial reasoning. For each failure scenario $i$, the model generates a dynamic recovery tuple $\hat{d}_i = \langle \hat{t}_{stop}, \hat{t}_{rb}, \mathbf{v}_i \rangle$, consisting of the **Stop Time** $\hat{t}_{stop} \in \mathbb{R}_{\geq 0}$ representing the predicted timestamp for operation cessation upon failure detection, the **Rollback Time** $\hat{t}_{rb} \in \mathbb{R}_{\geq 0}$ specifying the target historical state for reversion, and the **3D Correction Vector** $\mathbf{v}_i = [v_x, v_y, v_z]^\top$. In this vector, each discrete element $v_a \in \{-2, \ldots, 2\}$ $(a \in \{x, y, z\})$ quantifies the spatial adjustment intensity required to compensate for the manifested execution error. To assess corrective capabilities of multimodal models in robotic manipulation, we design a dynamic evaluation framework in simulation, focusing on corrective behavior during task execution, measured by Geometric Correction Accuracy (GCA), Temporal Localization Accuracy (TLA), and Simulation Success Rate (SSR).

**VLM-VLA Self-Correction Mechanism** We propose a VLM-VLA closed-loop self-correction mechanism that addresses the limitations of conventional local repairs by correcting errors at their semantic and decision-making roots. Based on the defined state space, the recovery process is formulated as a *detection-rollback-correction* sequence. Given a historical trajectory $\tau = \{s_0, a_0, \ldots, s_{t_{stop}}\}$, the mechanism is defined as follows:

*Table 1.* Static Evaluation Results of Fault Type Classification (FTC), Sub-task Success Evaluation (SSE), and Fault Severity Rating (FSR) across VLA Models.[Keys: **Best**/**Second best** in group]Abbreviations and severity definitions are provided in Appendix E.

| Model | Fault Type Classification | | | | | | Sub-task Success Evaluation | | | | | Fault Severity Rating | | | |
|---|---|---|---|---|---|---|---|---|---|---|---|---|---|---|---|
| | SOE | LMA | LGD | TLE | TRM | VDF | P&P | SG | AOCI | MLT | FIH | L-0 | L-1 | L-2 | L-3 |
| Gemini-3-Flash(DeepMind, 2025a) | 0.2035 | **0.4215** | **0.2542** | **0.6602** | 0.1004 | **0.9074** | **0.6139** | 0.5704 | **0.5403** | **0.6604** | **0.6094** | **0.6956** | 0.1206 | 0.3783 | **0.7165** |
| GPT-5.2(OpenAI, 2025b) | 0.2516 | 0.2727 | 0.2226 | 0.3844 | 0.0633 | 0.3195 | **0.6433** | 0.6161 | **0.5530** | **0.7060** | 0.4906 | 0.2960 | 0.1580 | **0.5052** | 0.5632 |
| Llama-3.2-90B-Vision(Meta, 2024) | 0.2713 | 0.2400 | 0.1484 | 0.4691 | 0.1277 | **0.7383** | 0.5124 | **0.6203** | 0.4747 | 0.4940 | 0.5103 | 0.1987 | 0.4886 | 0.1691 | 0.2605 |
| Qwen3-VL-235B(Bai et al., 2025) | **0.5033** | 0.2776 | 0.0123 | **0.5483** | 0.0611 | 0.4002 | 0.4149 | 0.5751 | 0.4597 | 0.4919 | 0.4763 | 0.2490 | 0.5073 | 0.0087 | 0.4764 |
| InternVL2.5-78B(OpenGVLab, 2024) | 0.3786 | 0.1009 | 0.2329 | 0.4571 | 0.0404 | 0.4528 | 0.4092 | 0.5651 | 0.4614 | 0.4974 | **0.5853** | 0.3002 | 0.6861 | 0.0671 | 0.1405 |
| InternVL3-78B (OpenGVLab, 2025) | 0.4551 | 0.2078 | 0.0426 | 0.3692 | 0.0939 | 0.4460 | 0.4075 | 0.5236 | 0.4777 | 0.4315 | 0.5347 | 0.3065 | 0.6611 | 0.0288 | 0.2516 |
| Ovis2-34B(Lu et al., 2024) | 0.4770 | 0.2225 | 0.0084 | 0.4365 | 0.0437 | 0.3619 | 0.4343 | **0.6218** | 0.5234 | 0.4813 | 0.4257 | 0.0000 | **0.8669** | 0.2006 | 0.0511 |
| Doubao-Seed-1.6(Doubao Team, 2025) | 0.2684 | **0.2973** | **0.2401** | 0.2165 | 0.0526 | 0.3755 | 0.5022 | 0.4611 | 0.3867 | 0.4973 | 0.5082 | 0.6288 | 0.3371 | 0.1485 | 0.2141 |
| InternVL3-38B (Zhu et al., 2025) | 0.1751 | 0.0275 | 0.0381 | 0.4191 | 0.1004 | 0.2702 | 0.4868 | 0.5530 | 0.4597 | 0.4366 | 0.4882 | 0.3431 | 0.3576 | 0.3867 | 0.2427 |
| Doubao-1-5-vision-pro (Guo et al., 2025) | 0.2797 | 0.1859 | 0.0336 | 0.4757 | 0.0552 | 0.3534 | 0.3771 | 0.4170 | 0.4661 | 0.5294 | 0.4462 | 0.5236 | 0.3258 | 0.0013 | 0.2375 |
| InternVL3-14B (Zhu et al., 2025) | 0.4595 | 0.2167 | 0.0477 | 0.1965 | 0.0688 | 0.2481 | 0.3765 | 0.3960 | 0.4262 | 0.4600 | 0.5579 | 0.2469 | **0.7505** | 0.0087 | 0.0690 |
| Qwen2.5-VL-72B(Wu et al., 2025a) | **0.5602** | 0.1508 | 0.0045 | 0.2888 | **0.2183** | 0.0875 | 0.3602 | 0.3550 | 0.4124 | 0.4004 | 0.3180 | 0.0021 | 0.3243 | 0.2415 | 0.3091 |
| Phi-4(Phi Team, 2025) | 0.1299 | 0.0010 | 0.0000 | 0.0000 | **0.1854** | 0.0000 | 0.2795 | 0.3330 | 0.4046 | 0.3272 | 0.3742 | **0.9960** | 0.0000 | 0.0000 | 0.0175 |
| mPLUG-Owl3-7B (Ye et al., 2024) | 0.1685 | 0.0003 | 0.0000 | 0.0000 | 0.0142 | 0.0136 | 0.3164 | 0.3272 | 0.3684 | 0.3068 | 0.3049 | 0.2793 | 0.6154 | 0.0456 | 0.0166 |
| LLaVA-OneVision(An et al., 2025) | 0.0613 | 0.0000 | 0.0000 | 0.0000 | 0.0207 | 0.0000 | 0.2259 | 0.2836 | 0.2654 | 0.2736 | 0.2682 | 0.0931 | 0.0000 | **0.5031** | **0.5709** |

**1) Failure Detection and Stop:** The system identifies a failure and triggers a stop at the predicted timestamp $t_{stop}$ when the current state $s_{t_{stop}}$ falls into the failure state space $\mathcal{C}_{fail}$. Formally, the termination condition is expressed as: $s_{t_{stop}} \in \mathcal{C}_{fail}$. The robot ceases operation immediately upon detection, providing the temporal basis for the subsequent rollback operation.

**2) Rollback Step:** A rollback operator $\mathcal{R}$ is invoked to revert the robot from the failed state $s_{t_{stop}}$ to a previously recorded safe state $s_{t_{rb}}$:

$$\mathcal{R}(s_{t_{stop}}, t_{rb}) \doteq s_{t_{rb}}, \quad \text{s.t. } 0 \leq t_{rb} < t_{stop} \quad (1)$$

This ensures the recovery starts from a stable configuration within the execution history.

**Design Rationale of the Repair Interface:** The defined dynamic tuple—Stop Time, Rollback Time, and 3D Correction Vector—is an intentionally minimal yet sufficient interface to translate VLM diagnostic capabilities into actionable recovery signals. This design follows three principles: (1) clarity and simplicity, allowing general black-box VLMs to integrate without model-specific output heads or adapters; (2) direct mapping from semantic fault judgment through temporal localization to geometric correction, forming an interpretable detect–rollback–correct chain; (3) minimal sufficiency, providing the controller with unambiguous information necessary for effective execution. While richer representations such as bounding boxes, points, masks, or full trajectory sequences were considered, they introduce additional barriers: structured geometric annotations are not reliably producible by all VLMs, and translating them into executable corrections imposes a separate translation

bottleneck. Our goal is therefore not to maximize representational richness, but to balance generality, executability, and cross-model comparability. Experimental results validate this choice: even with this minimal interface, human-in-the-loop corrections yield 13% improvement in simulation and 35% on real robots, and precise rollback outperforms coarse task restarts, demonstrating that the interface is sufficient to support effective recovery while current VLMs remain limited by temporal and spatial grounding rather than interface expressivity.

**3) Spatial Correction:** Upon reaching $s_{t_{rb}}$, the system applies the 3D correction vector $\mathbf{v}$ to adjust the subsequent execution. The rectified initial action $a'_{t_{rb}}$ is computed as:

$$a'_{\hat{t}_{rb}} = \pi(s_{t_{rb}}) + \Delta\mathbf{p}, \quad \text{where } \Delta\mathbf{p} = \delta \cdot \mathbf{v} \quad (2)$$

Here, $\Delta\mathbf{p} = [\Delta p_x, \Delta p_y, \Delta p_z]^\top$ represents the physical spatial offset, and $\delta$ is the unit magnitude constant.

**Evaluation Capability Dimensions** The dynamic evaluation framework comprises three core capability dimensions: **Geometric Correction Accuracy (GCA).** GCA evaluates the model's ability to correct execution deviations along three aspects: Axis Alignment (AA), Direction Consistency (DC), and Magnitude Accuracy (MA). By analyzing spatial vectors of corrective actions, this metric quantifies whether the proposed corrections align with the intended task direction and scale.
**Temporal Localization Accuracy (TLA).** TLA formalizes the temporal reasoning required for rollback-based recovery. It defines two critical decision points: the Fault Onset Time (FOT), marking the earliest detectable deviation from successful execution, and the Rollback Initiation Time (RIT), indicating when the system should revert to a previous execution state. By requiring models to localize both moments,

*Table 2.* Simulation Results on Dynamic Evaluation Metrics: Geometric Correction Accuracy (GCA), Temporal Localization Accuracy (TLA), and Simulation Success Rate (SSR). [Keys: **Best**/**Second best** in group]

| Metrics | GCA | | | | | | | | | TLA | | SSR | | | |
| --- | --- | --- | --- | --- | --- | --- | --- | --- | --- | --- | --- | --- | --- | --- | --- |
| | X | | | Y | | | Z | | | FOT | RIT | 10 | object | goal | spatial |
| | AA | DC | MA | AA | DC | MA | AA | DC | MA | LFS | EFS | | | | |
| Human | 100.00 | 100.00 | 100.00 | 100.00 | 100.00 | 100.00 | 100.00 | 100.00 | 100.00 | 100.00 | 100.00 | 62.50 | 96.50 | 89.50 | 92.00 |
| Doubao-Seed-1.6 (Doubao Team, 2025) | 16.28 | 10.47 | 5.81 | **61.22** | 27.55 | 16.33 | **68.75** | **56.25** | **30.21** | 43.31 | **29.94** | 37.50 | **88.00** | 65.00 | 60.50 |
| GPT-5.2 (OpenAI, 2025b) | 45.35 | 24.42 | **15.12** | **63.27** | **32.65** | **20.41** | 33.33 | 22.92 | 11.46 | 40.13 | 19.75 | 37.00 | 70.50 | 51.50 | 48.50 |
| Gemini-2.5-Pro (DeepMind, 2025b) | 41.86 | 24.42 | 11.63 | 46.94 | **30.61** | 15.31 | 31.25 | 16.67 | 10.42 | **48.41** | **35.67** | 31.50 | 79.50 | 64.00 | 67.00 |
| Gemini-2.5-Flash (DeepMind, 2025b) | 41.86 | 19.77 | 11.63 | 38.78 | 22.45 | **17.35** | **36.46** | **28.12** | **13.54** | 31.21 | **29.94** | 37.00 | 81.50 | 66.50 | 70.00 |
| ChatGPT-4o-Latest (OpenAI, 2024) | **58.14** | **31.40** | **17.44** | 29.59 | 15.31 | 9.18 | 14.58 | 9.38 | 4.17 | 30.57 | 26.11 | 31.50 | 75.00 | 54.50 | 49.50 |
| GPT-5 (OpenAI, 2025a) | **68.60** | **32.56** | 12.79 | 21.43 | 12.24 | 7.14 | 8.33 | 4.17 | 0.00 | 31.85 | 27.39 | 36.00 | **86.00** | 55.00 | 52.00 |
| Llama-3.2-90B-Vision (Meta, 2024) | 12.79 | 4.65 | 1.16 | 14.29 | 8.16 | 5.10 | 31.25 | 12.50 | 5.21 | **52.87** | 23.57 | 29.50 | 56.50 | 66.50 | 70.00 |
| InternVL3-38B (Zhu et al., 2025) | 23.26 | 11.63 | 6.98 | 14.29 | 4.08 | 3.06 | 10.42 | 6.25 | 6.25 | 32.48 | 8.92 | 34.50 | 69.50 | 68.50 | 71.50 |
| Qwen-2.5-VL-72B (Wu et al., 2025a) | 20.93 | 11.63 | 8.14 | 1.02 | 0.00 | 0.00 | 16.67 | 10.42 | 5.21 | 14.01 | 5.10 | 28.00 | 66.00 | 74.50 | 79.00 |
| InternVL3-78B (OpenGVLab, 2025) | 12.79 | 3.49 | 1.16 | 12.24 | 4.08 | 2.04 | 8.33 | 6.25 | 3.12 | 29.30 | 5.10 | 35.00 | 71.00 | 65.50 | 72.50 |
| GPT-4o (OpenAI, 2024) | 4.65 | 2.33 | 1.16 | 0.00 | 0.00 | 0.00 | 18.75 | 9.38 | 3.12 | 34.39 | 14.01 | 27.50 | 70.50 | 56.50 | 66.00 |
| Qwen-2.5-VL-7B (Wu et al., 2025a) | 4.65 | 3.49 | 2.33 | 3.06 | 1.02 | 0.00 | 0.00 | 0.00 | 0.00 | 9.55 | 3.82 | 31.50 | 80.00 | **76.00** | 83.00 |
| Gemini-2-Flash (DeepMind, 2025b) | 0.00 | 0.00 | 0.00 | 0.00 | 0.00 | 0.00 | 1.04 | 1.04 | 0.00 | 3.18 | 2.55 | 40.00 | 85.50 | 75.50 | 82.50 |
| Qwen3-VL-235B (Bai et al., 2025) | 1.16 | 1.16 | 0.00 | 0.00 | 0.00 | 0.00 | 0.00 | 0.00 | 0.00 | 0.64 | 0.00 | **41.50** | 79.50 | **76.50** | **83.50** |
| Doubao-1-5-vision-pro (Guo et al., 2025) | 0.00 | 0.00 | 0.00 | 0.00 | 0.00 | 0.00 | 1.04 | 0.00 | 0.00 | 1.91 | 0.00 | 38.50 | 75.50 | **76.50** | **83.50** |
| OpenVLA-7B Only | - | - | - | - | - | - | - | - | - | - | - | **43.00** | **86.50** | **86.50** | **83.50** |

TLA captures whether a judger can support recovery policies that are temporally precise rather than relying on coarse task-level restarts alone.

**Simulation Success Rate (SSR).** SSR measures end-to-end task completion over the entire execution horizon in simulation, reflecting the overall stability and effectiveness of the closed-loop control system. SSR is computed over all four LIBERO task suites, providing a unified assessment of performance consistency across diverse manipulation settings.

## 4.3. Real-Time Evaluation

To evaluate the practical performance of multimodal models in real-world robotic manipulation, we conduct on-robot experiments. Video streams are transmitted to the VLM at 1 Hz for real-time fault diagnosis, rollback decisions, and corrective action recommendations.

## 4.4. Metrics

**Static Evaluation Metrics.** We employ the following three metrics to quantitatively assess diagnostic proficiency:

**1) FTC Accuracy :** This metric measures model precision in identifying the correct failure category:

$$\text{Acc}_{\text{FTC}} = \frac{1}{N} \sum_{i=1}^{N} \mathbf{1}(f_i = f_i^*) \tag{3}$$

where $N$ is the number of instances, $f_i^*$ is the ground truth fault type and $\mathbf{1}(\cdot)$ is the indicator function.

**2) Level-specific Severity Accuracy ($Acc_{FSR}^k$):** To investigate model sensitivity to failures of varying intensities,

we calculate the diagnostic accuracy independently for each severity level $k \in \{1, 2, 3\}$. The accuracy for level $k$ is defined as:

$$Acc_{FSR}^k = \frac{1}{N_k} \sum_{i \in \mathcal{D}_k} \mathbf{1}(l_i = l_i^*) \tag{4}$$

where $\mathcal{D}_k = \{i \mid l_i^* = k\}$ is the subset of samples with ground truth severity level $k$, and $N_k = |\mathcal{D}_k|$ is the total number of samples in that subset. This categorical breakdown allows us to identify potential *sensitivity gaps*, such as a model inability to distinguish critical Level-3 failures from minor Level-1 perturbations, which is crucial for assessing safety-critical performance.

**3) Subtask Success Accuracy (SSA):** This metric evaluates the ability of model to track task progress across multiple stages in practice:

$$SSA = \frac{1}{N} \sum_{i=1}^{N} \left( \frac{1}{m_i} \sum_{j=1}^{m_i} \mathbf{1}(s_{i,j} = s_{i,j}^*) \right) \tag{5}$$

where $m_i$ is the number of subtasks in sample $i$, and $s_{i,j}^*$ is the success label for the $j$-th subtask.

**Dynamic evaluation metrics.**

**1) Geometric Correction Quality (GCQ):** The model predicts a 3D correction vector $\mathbf{v} = [v_x, v_y, v_z]^\top$, where $v_a = 0$ signifies no correction. For each axis $a$ requiring a ground truth correction ($v_a^* \neq 0$), we evaluate performance through three hierarchical binary indicators: (i) **Axis Alignment (AA)**, $\mathbf{1}_{AA}^{(a)} = \mathbf{1}(v_a \neq 0)$, which as-

sesses whether the model identifies the necessity of correction on the specific axis; (ii) **Direction Consistency (DC)**, $\mathbf{1}_{DC}^{(a)} = \mathbf{1}_{AA}^{(a)} \cdot \mathbf{1}(\text{sgn}(v_a) = \text{sgn}(v_a^*))$, ensuring the corrective force is applied in the correct direction; and (iii) **Magnitude Accuracy (MA)**, $\mathbf{1}_{MA}^{(a)} = \mathbf{1}_{DC}^{(a)} \cdot \mathbf{1}(v_a = v_a^*)$, representing a precise match in correction intensity. This hierarchical formulation ensures that higher-level metrics (DC and MA) are strictly contingent upon the success of the preceding diagnostic steps in order.

**2) Temporal Localization Accuracy (TLA):** To evaluate model temporal responsiveness, we define binary scoring functions for **Stop Time** ($t_{\text{stop}}$) and **Rollback Time** ($t_{\text{rb}}$) predictions, measuring whether the predicted timestamps fall within a tolerance window around the ground truth.

The score for stop time prediction $\hat{t}_{stop}$ is defined as:

$$S_{\text{stop}} = \begin{cases} 1, & \text{if } \hat{t}_{\text{stop}} \in \left[t_{\text{stop}}^* - \Delta_{\text{stop}}^-, \ t_{\text{stop}}^* + \Delta_{\text{stop}}^+\right] \\ 0, & \text{otherwise} \end{cases} \quad (6)$$

$\Delta_{\text{stop}}^-$ and $\Delta_{\text{stop}}^+$ denote the early and late tolerances for stop-time prediction. This asymmetric window allows for a slight delay in detection while penalizing premature stops.

Similarly, the score for rollback time prediction $\hat{t}_{rb}$ is constrained by a tighter window to ensure precise recovery:

$$S_{\text{rb}} = \begin{cases} 1, & \text{if } \hat{t}_{\text{rb}} \in \left[t_{\text{rb}}^* - \Delta_{\text{rb}}^-, \ t_{\text{rb}}^* + \Delta_{\text{rb}}^+\right] \\ 0, & \text{otherwise} \end{cases} \quad (7)$$

where $t^*$ denotes the GT timestamp, and $\Delta_{\text{rb}}^-$ and $\Delta_{\text{rb}}^+$ are the early and late tolerances for rollback-time prediction.

**Real-time evaluation metrics** To assess on-robot fault diagnosis performance, we adopt metrics that jointly measure sensitivity, robustness, and task effectiveness. Recall quantifies the proportion of true faults correctly detected, emphasizing sensitivity to failures where missed detections incur high cost. Precision measures the proportion of predicted faults that are correct, reflecting reliability and resistance to false alarms. The F2-Score combines Recall and Precision with greater emphasis on Recall, providing an integrated assessment of fault detection capability. The False Positive Rate (FPR) captures the proportion of non-fault instances incorrectly flagged as faults, indicating intervention stability. Finally, Success Rate (SR) directly measures task completion under model assistance, serving as the most direct indicator of real-world operational effectiveness.

## 5. Results

### 5.1. Experimental Setup

**Data Acquisition.** For simulation, we fine-tuned OpenVLA-7B (Kim et al., 2024) using data from the LIBERO bench-

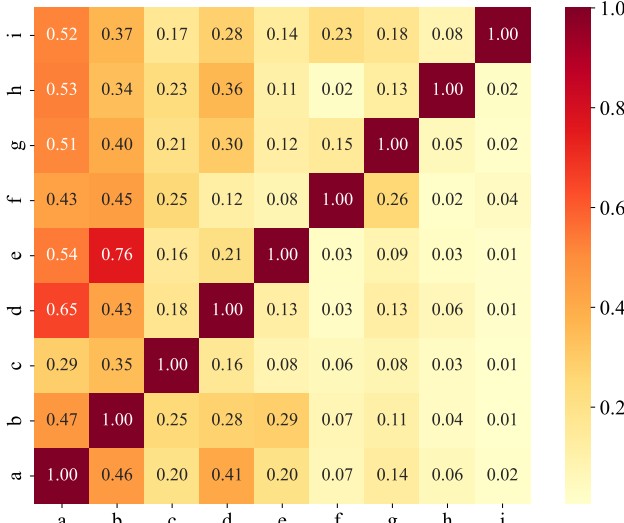

*Figure 4.* **Faults correlation analysis**. Conditional failure dependency heatmap over fine-grained fault types. Each entry denotes $P(b \mid a)$, the probability that failure type $b$ occurs simultaneously or subsequently given failure type $a$.

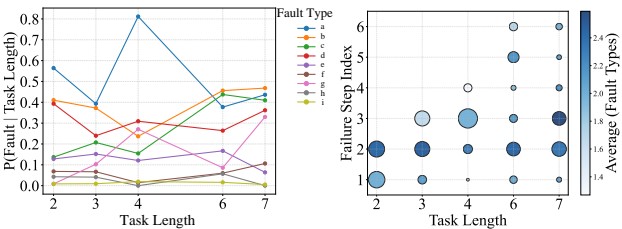

*Figure 5.* The left figure shows the relationship between different fault types occurrence probability and task length, while the right figure shows the relationship between the fault-stage occurrence probability and task length.

mark (Liu et al., 2023), and conducted simulation evaluations on both the LIBERO and LIBERO-Pro (Zhou et al., 2025) task suites. Then we inject systematic perturbations into actions and visual inputs to elicit diverse failures in spatial reasoning and multi-step execution. Real-world samples were acquired via a Franka arm across *Tea Preparation* and *Charging Plug Insertion* tasks, where variations in lighting, distractors, and target poses induced a wide range of varied execution faults.

**Curation and Benchmarking.** The resulting dataset is curated with fine-grained annotations (failure type, subtask, and severity). We evaluated a diverse model suite, including GPT-4o, Gemini-2.5, and Qwen series, across four simulation subsets and physically realistic environments.

### 5.2. Static Evaluation Results

Static evaluation (Table 1) reveals a clear performance stratification among VLMs. Proprietary models, notably Gemini-3-Flash and GPT-5.2, consistently dominate the leaderboard: Gemini excels in perception-centric tasks (VDF: 0.9074;

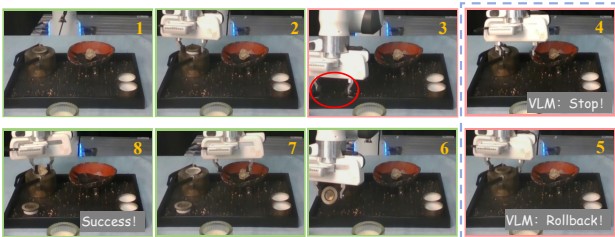

*Figure 6.* Failure recovery in the **real-robot** make-tea task.

*Table 3.* **Real-robot evaluation results**. The performance is measured across diagnostic metrics (Recall, Precision, F2-Score, and FPR) and the manipulation Success Rate (SR). [Keys: **Best**/**Second best**]

| Model | Recall | Precision | F2-Score | FPR | SR |
|---|---|---|---|---|---|
| Gemini-2.5-Pro (DeepMind, 2025b) | 0.1795 | 0.4667 | 0.2047 | 0.1333 | **20** |
| Gemini-2.5-Flash (DeepMind, 2025b) | 0.1429 | 0.4167 | 0.1645 | **0.0933** | **25** |
| Llama-3.2-90B-Vision (Meta, 2024) | 0.4821 | 0.6429 | **0.5075** | 0.4286 | 0 |
| GPT-5-2 (OpenAI, 2025b) | **0.8571** | 0.4286 | **0.7143** | 0.7568 | 0 |
| Qwen-2.5-VL-7B (Wu et al., 2025a) | **0.5333** | 0.2667 | 0.4444 | 0.7097 | 10 |
| Qwen3-VL-235B (Bai et al., 2025) | 0.2162 | **0.7273** | 0.2516 | **0.0588** | 15 |
| Doubao-1-5-vision-pro(Guo et al., 2025) | 0.1923 | **0.6667** | 0.2242 | 0.1042 | **20** |
| Doubao-Seed-1.6 (Doubao Team, 2025) | 0.2353 | **0.6667** | 0.2703 | 0.1395 | 15 |
| Groot-N1.5-3B Only (NVIDIA et al., 2025) | - | - | - | - | 30 |
| Human | - | - | - | - | 65 |

L-3: 0.7165), whereas GPT-5.2 exhibits superior robustness in long-horizon reasoning, leading the multi-object long-sequence tasks (MLT: 0.7060). Conversely, open-source models like Qwen2.5/3-VL and InternVL achieve parity in high-level semantic understanding, with Qwen2.5-VL-78B outperforming all closed-source models in spatial object errors (SOE: 0.5602). Across fault categories, a sharp perceptual-logical divergence emerges. While visual distortion faults are handled reliably, performance collapses on task relevance misjudgment (TRM) and lack of goal-directedness (LGD), with even top-tier models failing to surpass 0.2 accuracy. A similar pattern persists in severity assessment: models exhibit distinct sensitivities toward minor (L-1) or critical (L-3) failures, yet moderate-level faults (L-2) remain a universal blind spot (¡0.2). Interestingly, perception-limited models like LLaVA-OneVision maintain strong critical failure detection (L-3: 0.5709), suggesting their utility as conservative *safety sentries* rather than comprehensive diagnosticians.

Ultimately, SSE is the most discriminative signal for assessing a model's understanding of task-level execution outcomes. In contrast, FTC primarily reflects a model's multimodal perception and semantic classification capability. Together, these results indicate that the key limitation of current diagnostic agents lies not in fault recognition itself, but in reliable spatio-temporal grounding of task progress and outcomes, motivating the need for more finer-grained supervision in future VLA systems.

## 5.3. Dynamic Evaluation Results

Dynamic evaluation measures the synergy between diagnostic reasoning and physical execution. GCA results reveal a universal difficulty hierarchy across all models: Magnitude Accuracy (MA) ¿Direction Consistency (DC) ¿Axis Alignment (AA). While top-tier models like GPT-5 and GPT-5.2 demonstrate superior diagnostic precision, achieving the highest AA scores (up to 68.60%).Their Magnitude Accuracy (MA) remains a significant bottleneck (¡20%), indicating a persistent gap in fine-grained geometric control. We observe clear and consistent performance differences across X, Y, and Z. Y achieves the most balanced and overall strongest performance across AA, DC, and MA, while X remains competitive on AA and DC but underperforms on MA. In contrast, Z exhibits a pronounced degradation across all metrics, particularly MA, indicating a substantially more challenging setting. This trend is consistent across models, suggesting that the observed gaps are driven by intrinsic differences among the evaluation settings rather than model-specific effects.

TLA results further distinguish models by their temporal responsiveness, with Llama-3.2-90B and Gemini-2.5-Pro leading in fault onset and rollback timing. However, high diagnostic scores do not linearly translate to task success. SSR results expose a *correction paradox*: aggressive models (e.g., GPT series) often suffer from lower success rates than the baseline due to secondary failures induced by imprecise adjustments. In contrast, conservative models like Qwen3-235B maintain high SSR (83.50%) primarily through a *silent intervention* strategy, underscoring the critical trade-off between proactive correction and operational stability in real-world deployments.

## 5.4. Real-Time Evaluation Results

Real-robot evaluations (Table 3) validate the *sensitivity-stability paradox*. GPT-5-2 achieves the highest sensitivity (Recall: 0.8571, F2-Score: 0.7143), but its high FPR (0.7568) causes task failure (SR: 0), indicating that over-sensitive diagnosis can disrupt nominal executions. By contrast, Qwen3-VL-235B is more conservative, with the highest Precision (0.7273) and lowest FPR (0.0588), though its SR (15%) remains below the Groot-N1.5-only baseline. The drop in real-robot success rate reflects two competing effects: *correction benefit* and *intervention cost*. Current VLMs may misclassify trajectories that would have succeeded without intervention, triggering unnecessary rollback or correction and turning successes into failures. However, in several cases, they correctly detect that the VLA policy has entered a local optimum or erroneous trajectory, trigger rollback, and enable successful re-execution, converting failed trials into successful ones. These cases suggest that the proposed detect–rollback–correct chain can translate correct diagnosis into effective physical recovery. Existing

VLMs can answer diagnostic questions in static settings, but this ability does not reliably transfer to real robots, where recovery requires spatio-temporal grounding, calibrated correction, and confidence-aware decisions. Thus, our real-robot experiments do not claim that off-the-shelf VLMs solve robotic recovery; rather, they expose the mismatch between current VLM benchmarks and the capabilities needed to improve embodied manipulation success. VLA-FixBench uses fault diagnosis and rollback-based recovery as a testbed to bridge this gap.

**Real-Time Latency and Inspection Protocol** Our real-robot evaluation uses a sparse *pause-and-inspect* protocol instead of querying the VLM at every control step. Each task typically triggers only 3–6 diagnoses. Between two inspection points, the robot is controlled solely by the VLA policy, so continuous execution is not affected by VLM latency. In each cycle, the VLA executes for a fixed 1-second window, recorded as a 20-FPS video segment; robot motion and VLA inference are then paused for VLM-based diagnosis; if a recoverable failure is detected, rollback and correction are performed before execution resumes. Thus, VLM latency is explicitly included in the closed-loop cycle:

$$T_{\text{loop}} = T_{\text{exec}} + T_{\text{vlm}} + T_{\text{rollback}}, \tag{8}$$

where $T_{\text{exec}} = 1\text{s}$, $T_{\text{vlm}}$ is the average VLM inference latency, and $T_{\text{rollback}}$ is the physical rollback and correction time. This design is not merely for latency accommodation: when the VLA enters an erroneous trajectory or local failure mode, it may repeat ineffective actions and consume more time than sparse VLM inspection. The real-robot setup is therefore a sparse diagnostic-and-recovery loop that trades limited inspection latency for recovery from failures that the VLA alone cannot escape.

### 5.5. Alignment Between Static and Dynamic Evaluation

Static evaluation primarily measures high-level fault understanding and reasoning, whereas simulation evaluation emphasizes spatiotemporal precision and physical execution accuracy. As a result, models excelling in semantic reasoning do not necessarily perform well in closed-loop control, and no existing model achieves strong performance across both dimensions. We compute rank correlations between static and simulation evaluations, obtaining Spearman's $\rho = 0.43$ and Kendall's $\tau = 0.33$. This underscores the necessity of maintaining both evaluation subsets to comprehensively assess VLA robustness.

### 5.6. Ablation Study

**Effectiveness of Self-Correction** Across all evaluated tasks, human-in-the-loop correction consistently yields substantial performance gains over open-loop execution, improving average success rates by 13% points in simulation and by 35% points on real robots , significantly enhancing task robustness in both simulated and physical environments. These results demonstrate that closed-loop self-correction is critical for mitigating error accumulation and enabling reliable long-horizon manipulation.

**Impact of Rollback Step Selection** We further analyze recovery strategies by comparing naive task-level rollback with rollback to intermediate execution steps. While resetting to the initial task pose improves performance over no correction, it remains consistently inferior to rollback strategies that localize failures to appropriate temporal segments, achieving an average improvement of 8.3% points in simulation. This highlights that effective recovery requires precise temporal fault localization rather than coarse task restarts, validating the design of our rollback mechanism.

## 6. Conclusion and Outlook

We present VLA-FixBench, a unified benchmark for evaluating VLM-driven fault diagnosis and recovery in robotic manipulation, integrating static reasoning, simulation-based closed-loop testing, and real-robot validation. We further show that VLM–VLA collaboration with closed-loop self-correction improves robustness on average. Looking ahead, our results suggest that the main bottleneck is not merely static fault recognition, but the reliable transformation of semantic diagnosis into temporally grounded, spatially calibrated, and physically executable recovery actions. This highlights the need for finer-grained supervision, safer correction policies, confidence-calibrated intervention mechanisms, and stronger spatio-temporal grounding in future VLM–VLA systems.

## Acknowledgements

a) This work was supported by New Generation Artificial Intelligence-National Science and Technology Major Project (2025ZD0124104) in collaboration with Shanghai Artificial Intelligence Laboratory. b) This work was supported by the Shanghai Municipal Special Program for Basic Research on General AI Foundation Models (Grant No. 2025SHZDZX025D09), in collaboration with Shanghai Artificial Intelligence Laboratory. c) This work was supported in part by the National Natural Science Foundation of China under Grants 62225112, 625B2118.

## Impact Statement

This paper studies failure diagnosis and recovery for Vision-Language-Action (VLA) models. By analyzing how VLM-based judgers detect and respond to manipulation failures, our work aims to improve robotic reliability and reduce unintended behaviors in long-horizon tasks. These mechanisms may help reduce physical damage and operational risks in structured environments such as warehouses, laboratories, and assistive settings.

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

# A. Dataset Overview and Composition

**Dataset Statistics.** The dataset comprises 6,034 task execution episodes collected from both simulated and real-robot settings. It includes 40 simulated task settings and two real-robot task settings, namely Tea Preparation and Charging Plug Insertion. These settings cover pick-and-place, multi-step composite manipulation, switch/button interaction, and simple tool-use scenarios. Pick-and-place tasks constitute roughly 45% of samples, multi-step composite tasks 35%, and the remaining task types about 20%, ensuring diversity in task complexity, operational patterns, and scene coverage.

The dataset systematically captures a wide spectrum of failure types, stages, and severity levels. Frequent failures such as *Jitter*, *Grasp Instability*, and *Stagnation* appear in over 25% of samples, while rare failures (e.g., *Wrong Placement*, *Target Occluded*) occur in less than 7%, supporting edge-case analysis. Approximately 64% of episodes contain 2–3 failure types, and 7.5% include four or more, allowing multi-failure interaction studies. In terms of severity, level-2 (self-recoverable) failures dominate at 63.2%, followed by level-3 failures at 13%, while error-free executions account for 16%. Temporal analysis indicates that failures are evenly distributed in short tasks but concentrate in middle-to-late stages for longer tasks, emphasizing the importance of robustness in long-horizon manipulations.

0. **No Error / Success** – Task completed smoothly without any issues.

1. **Jitter** – High-frequency shaking or vibration of the arm/gripper during motion or hold.

2. **Grasp Instability / End-effector Error** – Incorrect positioning or unstable grasp (slip, off-center hold).

3. **Stagnation / Local Minima** – Robot stops or loops without progress.

4. **Aimless Wandering** – Motion without approaching the target.

5. **Sequence Error / False Success** – Skipping or misordering steps, or false assumption of success.

6. **Wrong Placement** – Correct manipulation but incorrect final pose/location.

7. **Target Occluded** – Target not visible to the robot.

8. **Task Object Accident** – Accidental dropping or knocking over of the target object.

9. **Non-Task Object Accident** – Collision with or displacement of non-target objects.

10. **Visual Perception Failure** – Blur, black screen, exposure issues, or noise in visual input.

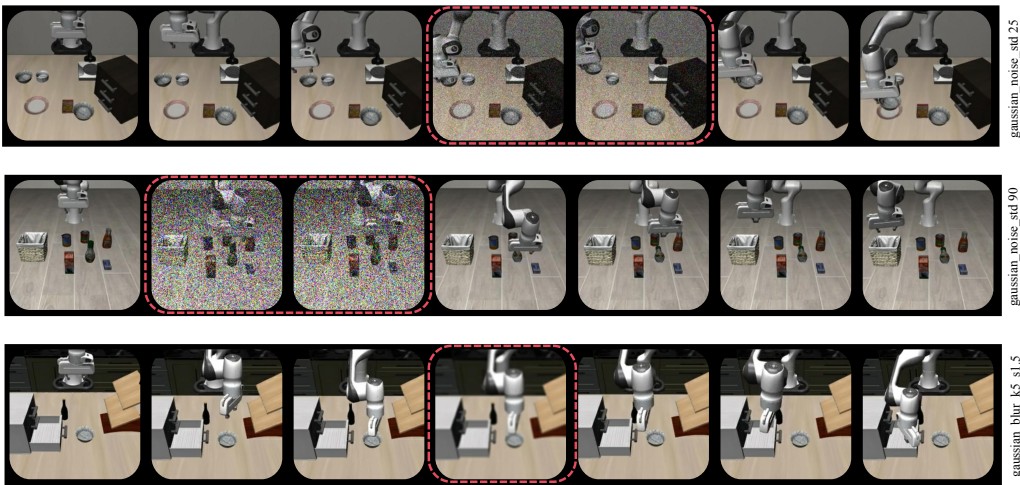

*Figure 7.* Examples of perception error types.

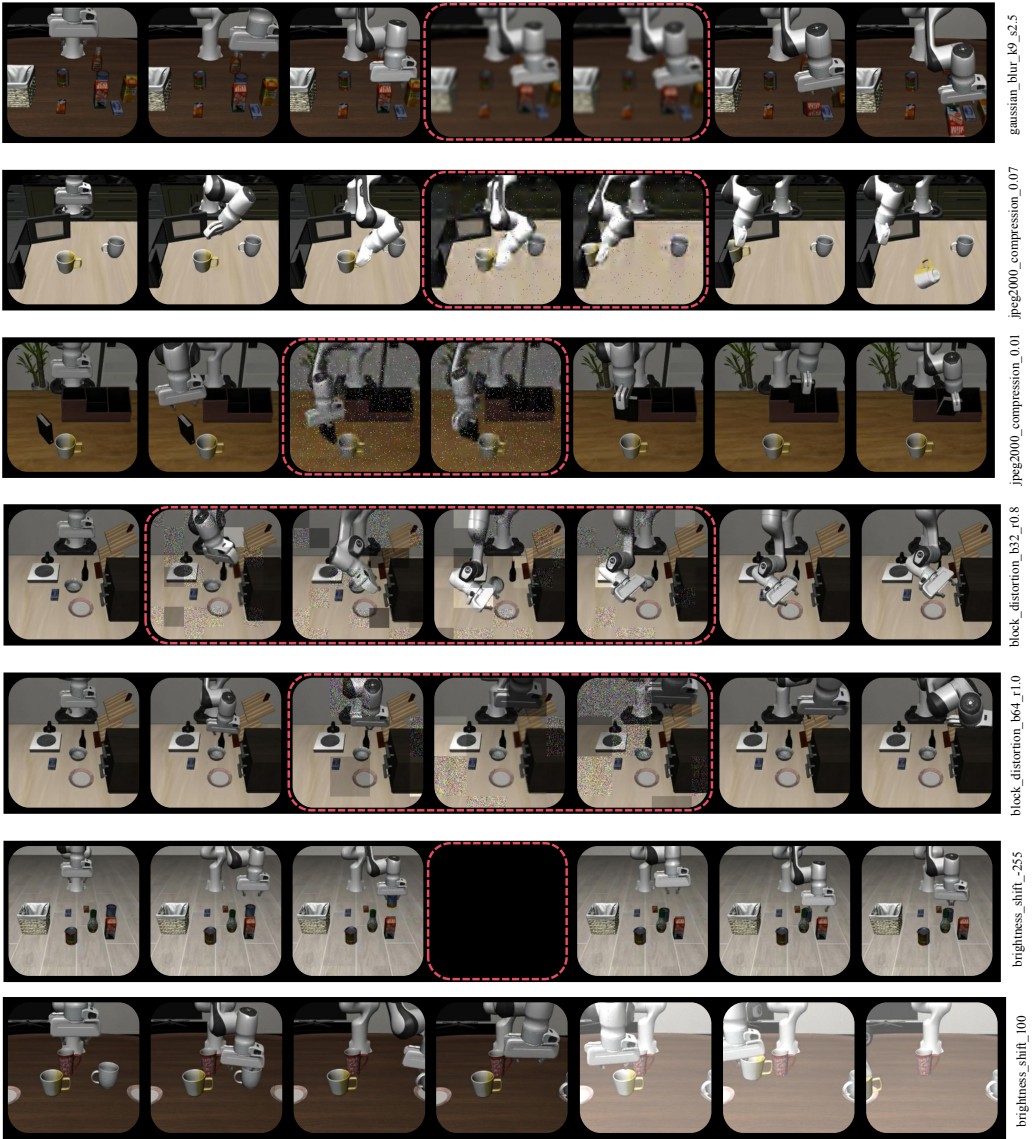

*Figure 8.* Examples of perception error types.

## B. Annotation

All human-involved procedures, including data annotation and human baseline testing, were conducted in accordance with institutional ethical guidelines. The annotators consisted of ten graduate students with engineering backgrounds, who were compensated fairly in accordance with institutional standards. No personally identifiable information (PII) was collected.

**Data Annotation Protocol**

We employ an video annotation tool to manually annotate errors in robot manipulation videos. The annotation process consists of four main stages: interface familiarization, video playback control, error-type annotation with phase identification, and result saving/navigation.

**Annotation Interface** The annotation interface, illustrated in Fig. 10, is divided into four functional regions:

- **Left Video Panel:** Displays the robot operation video for annotation, with controls for play, pause, and frame-by-frame navigation.

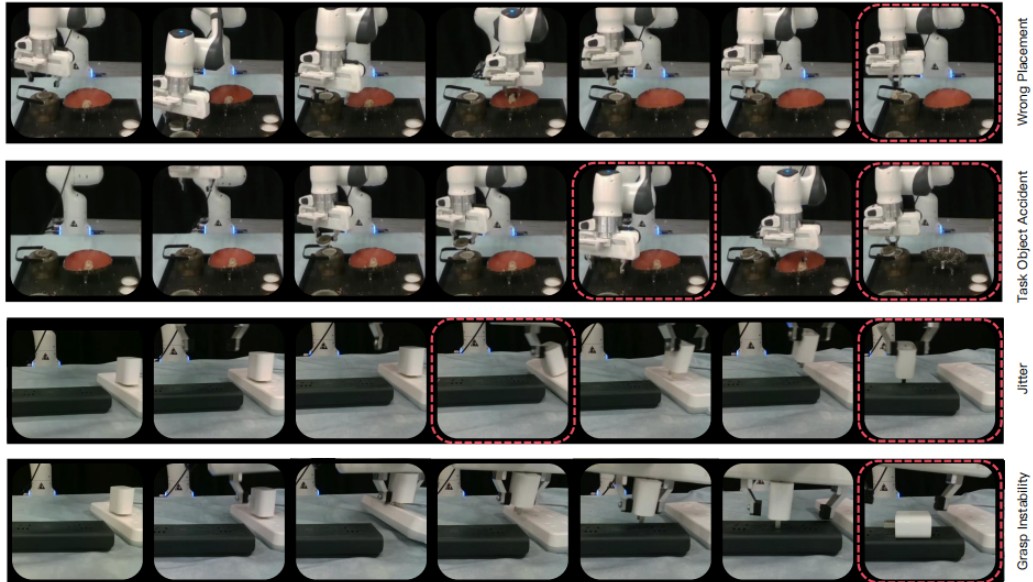

*Figure 9.* Examples of failure types in real robot experiments.

- **Central Annotation Panel:** Provides a set of checkboxes for error types. The annotator selects all observed error categories based on the video content.

- **Right Information Panel:** Shows metadata for the current video, including task difficulty level (Level 1–3), task ID, and success/failure indication.

- **Bottom Control Panel:** Contains buttons for saving annotations, navigating to the previous/next video, and jumping to a specific video by ID.

**Annotation Procedure** The standard workflow for annotators is as follows:

**(1) Video Playback & Observation.** The annotator first plays the video in the left panel to observe the complete task execution. To enhance efficiency, the following keyboard shortcuts are supported: **Key A:** Rewind video by 0.5 seconds. **Key S:** Toggle play/pause.**Key D:** Fast-forward video by 0.5 seconds.

**(2) Multi-label Error Annotation.** During or after playback, the annotator selects all applicable error types from the central panel. Our system defines error categories primarily as:**Interaction Errors:** Includes sliding without clear task progress, continuing after a failed grasp, incorrect phase sequence, misplacement, and complete occlusion of the target object.**Object State Errors:** Includes the task object being knocked over, dropped, or moved out of the workspace.

**(3) Phase Identification & Temporal Localization.** For complex tasks requiring fine-grained analysis, the annotator must specify the phase during which the error occurs and mark its start and end timestamps on the video timeline to ensure precise error localization.

**(4) Saving & Navigation.** Upon completion, clicking *"Save & Next"* submits the annotation and loads the next video. Annotators can click *"Previous"* to review or modify the last record, or directly navigate to a specific video by entering its ID and clicking *"Jump"*.

**Quality Assurance** All annotators undergo standardized training using the same guideline documentation. Prior to formal annotation, each annotator must complete a set of calibration videos, achieving an accuracy of over 90% to qualify. To ensure reliability, 10% of the annotated videos are randomly selected and reviewed by an expert annotator.

## C. Real - Metrics

**Fault Sensitivity and Coverage.** Measured by Recall, this metric reflects the model's ability to detect faults during real robot execution. Higher Recall indicates broader fault coverage, potentially at the cost of increased false alarms.

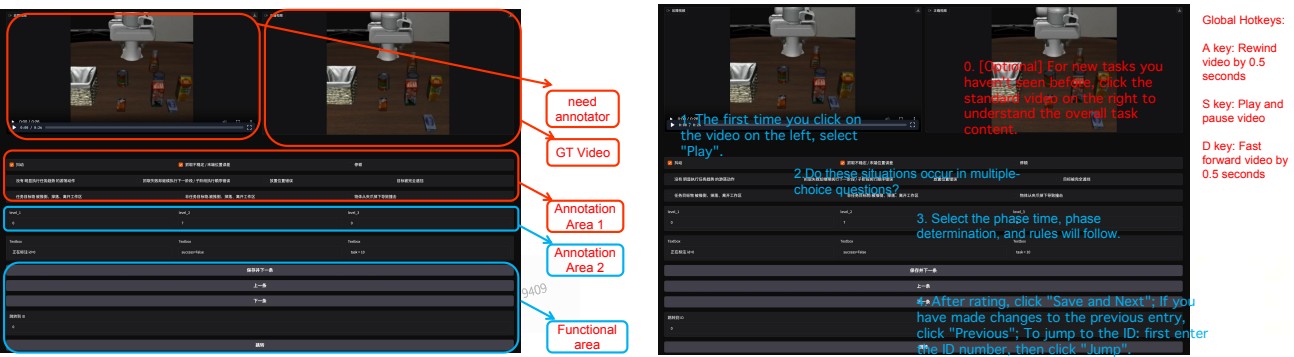

*Figure 10.* Annotation interface and operating procedure used for VLA-FixBench data annotation.

**Prediction Accuracy and Robustness.** Precision and FPR evaluate the correctness of fault predictions and the frequency of false interventions. High Precision together with low FPR indicates robust and reliable decision-making.

**Overall Fault Diagnosis Performance.** The F2-Score balances Recall and Precision with greater emphasis on Recall, providing an integrated measure of fault detection capability.

## D. Tolerance Windows for Temporal Scoring

For the binary temporal scoring functions in Sec. 4.4, we use fixed asymmetric tolerance windows. Specifically, the stop-time score $S_{\text{stop}}$ is evaluated within

$$\left[t_{\text{stop}}^* - \Delta_{\text{stop}}^-, \ t_{\text{stop}}^* + \Delta_{\text{stop}}^+\right],$$

where we set $\Delta_{\text{stop}}^- = 1$ and $\Delta_{\text{stop}}^+ = 3$ (in frames). This design penalizes premature stops more strongly while allowing a small delay in detection.

Similarly, the rollback-time score $S_{\text{rb}}$ is evaluated within

$$\left[t_{\text{rb}}^* - \Delta_{\text{rb}}^-, \ t_{\text{rb}}^* + \Delta_{\text{rb}}^+\right],$$

where we set $\Delta_{\text{rb}}^- = 2$ and $\Delta_{\text{rb}}^+ = 1$ (in frames). This tighter window encourages precise recovery timing.

## E. Statics Table

In the Stable-Static evaluation setting, the abbreviation labels describe both the semantic type of manipulation failures and the task domain in which each failure is observed. For fault-type categories, SOE denotes Spatial-Object Errors, referring to cases where the gripper or end-effector fails to reach the correct target position and therefore becomes spatially misaligned with the intended object or interaction point; LMA denotes Low-level Motion Anomaly, which captures abnormal low-level motion behaviors of the robot arm or gripper, including jittering, overshooting, oscillation, stagnation, or unstable holding; LGD denotes Lack of Goal-Directedness, where the robot may continue moving smoothly but fails to make meaningful progress toward the task goal, or remains trapped in local loops, idle states, or ineffective motion patterns; TLE denotes Temporal/Logical Error, referring to failures in sub-task ordering, skipped necessary steps, or continuing subsequent actions under the false assumption that a previous failed sub-task has succeeded; TRM denotes Task Relevance Misjudgment, where the robot misidentifies which object, region, or interaction target is relevant to the current instruction and consequently manipulates the wrong target or performs task-irrelevant behavior; and VDF denotes Visual Distortion Failures, covering failures caused by corrupted or incomplete visual input, such as image noise, blur, distortion, occlusion, over-exposure, under-exposure, or missing visual information. For task-domain categories, P & P denotes Pick-and-Place, covering standard manipulation tasks that involve grasping, transporting, and placing objects; SG denotes Spatial Grasping, emphasizing tasks that require accurate spatial reasoning for grasp pose selection and object approach; AOCI denotes Articulated Object and Container Interaction, covering constrained interactions with drawers, doors, microwave doors, containers, or other articulated structures; MLT denotes Multi-object Long-sequence Tasks, referring to long-horizon manipulation scenarios involving multiple objects, multiple sub-steps, and sequential reasoning; and FIH denotes Functional Interaction and Hybrid Tasks, which include fine-grained functional manipulation, compound skills, coordinated object use, or hybrid task settings

requiring deeper understanding of object usage and task purpose. Together, these Stable-Static labels characterize what type of failure occurs and under which manipulation domain the failure is observed.

## F. Dynamic Table

In the Stable-Dynamic evaluation setting, the abbreviation labels describe temporal fault localization and geometric correction quality during continuous robotic execution. For temporal localization, FOT denotes Fault Onset Time, the earliest time point at which the execution can be identified as deviating from a successful task trajectory; RIT denotes Rollback Initiation Time, defined as the target historical time point to which the system should roll back in order to return to a previously safe and recoverable state; LFS denotes Late Failure Step, a tolerant fault-detection metric that allows the predicted failure step to be slightly later than the true failure onset while penalizing overly premature stopping; and EFS denotes Earlier Failure Step, a tolerant rollback metric that allows the predicted rollback point to be slightly earlier than the ideal rollback target, since earlier rollback is often safer than rolling back too late. For geometric correction, AA denotes Axis Alignment, measuring whether the predicted correction is made along the same spatial axis as the ground-truth correction; DC denotes Direction Consistency, measuring whether the predicted correction follows the same direction as the ground-truth correction along the correct axis; and MA denotes Magnitude Accuracy, measuring whether the predicted correction matches the ground-truth correction in scale or magnitude. Together, these Stable-Dynamic metrics evaluate when a model identifies or rolls back from a failure, and whether its corrective action is geometrically aligned with the required recovery direction and magnitude.

