# OpenReview forum: "Can VLMs Diagnose and Recover from VLA Manipulation Faults?"
_ICML.cc/2026/Conference — ICML 2026 regular_

### Official Review · Reviewer_mVfz · 2026-03-07

**Soundness:** 3
**Presentation:** 3
**Significance:** 3
**Originality:** 3
**Overall Recommendation:** 4
**Confidence:** 5

**Summary:**

This paper studies failure corrections of robot policies with spatial understanding of VLMs.  It proposes a dataset that evaluates failure understanding across three dimensions, including classifying error types, rating fault severity and predicting subtask success. Additionally, this paper evaluates VLMs' capabilities of failure recovery in terms of corrective action prediction, and failure, rollback time detection.  With these benchmarks, this paper presents a systematic analysis, showing that existing VLMs carry individual inductive biases--each excels at only certain dimensions while struggling on others.  Notably, this paper raises a concern that the diagnostic performance does not fully reflect failure recovery preformance in real robotic systems.  As a result, how to bridge the evaluation gap should be addressed in future work.

**Compliance With Llm Reviewing Policy:**

Affirmed.

**Final Justification:**

The authors didn't address my concerns.  I'd maintain my original score.

**Key Questions For Authors:**

1. What are the definitions of SOE, LMA, LGD, TLE, TRM, VDF, P&P, SG, AOCI, MLT, FIH in Figure 1 and LFS, EFS in Figure 2? I understand some of these terms are mentioned in the Appendix, however, their descriptions are still unclear or confusing.
2. How do you justify the selected benchmark? Specifically, why Fault Type Classification, Fault Severity Rating, Subtask Success Evaluation, Stop Time Detection, Rollback Time Detection and 3D Correction Vector matter if diagnostic performance of these tasks does not translate to failure recovery on real robotic systems?
3. What are the connections between static and dynamic evaluation? In other words, why should one consider static evaluation if it doesn't correspond to real-time failure recovery?
4. How do you justify the prediction of rollback time? Is it reasonable to assume a reversible Markov Decision Process for robotic manipulation tasks?
5. What is the take-away message of this paper if the proposed diagnostic performance fails to translate to real-robot failure recovery?

**Limitations:**

Authors should discuss limitations and hidden assumptions of the selected benchmark.

**Strengths And Weaknesses:**

### Strengths

1. This paper offers a benchmark that thoroughly evaluates a VLM's capability in diagnosing and correcting failures of robot policies.  For the former, it considers classifcation of error types, rating the severity of action errors, and prediction of subtask sucess; for the latter, it considers prediction of residual actions and detection of time when failure occurs and the policy should rollback.  Additionally, this paper evaluates a wide range of VLMs (15 models), meanwhile, it conducts real-robot experiments to validate the proposed benchmark.  Therefore, this paper is technically sound in terms of methodolgy and experiments.
2. This paper did a good job in presenting experimental results and the overall framework.  The radar chart of figure 3 clearly shows that there's no winner VLM for robot policy diagnosis and recovery. The diagram of figure 1 clearly summarize the proposed benchmark.
3. I'll describe the significance of this paper in two distinctive aspects (see the Weaknesses section for the negative aspect). This paper presents a systematic study using a complete list of metrics to evaluate VLMs' diagnostic performance.  While the real-robot evaluation results are disappointing, they still offer a good lesson--discriminatively modeling failure understanding is likely to be an incorrect direction--for the community.
4. Given the depth and width of the proposed benchmark, I believe this paper has enough originiality.

---

### Weaknesses

1. Following S3, although this paper presents a thorough study in robotic failure understanding, the real-robot evaluation results are disappointing--the diagnostic performance does not reflect failure recovery preformance in real robotic systems.  Therefore, it is arguably true that this paper delivers little scientific significance as it offers no insight into robotic failure recovery.
2. While introducing an overly long list of metrics for static and dynamic evaluation, this paper fails to present numerical results clearly.  For example, Table 1 and 2 include multiple notations (e.g. SOE, TLE, LFS ...) without proper definition.  It's difficult to understand these tables.
3. This paper lacks proper justification on the selected benchmark.  For example, why predicting 3D correction vector, the stop time and the rollback time are appropriate for robotic failure recovery.  In fact, this design choice assumes the Markov Decision Process is revertible (in order to rollback to a specific time), which is not true in most robotic manipulation tasks like *pouring water into a mug*. Such limitations are not discussed in the main paper.

---

I have mixed rating of this paper.  On the one hand, I feel positive for this paper given its enormous engineering effort;  on the other hand, it's hard to draw constructive conclusion from this paper.  Therefore, I suggest weak accept for the initial review, but I'm also open to adjust the rating.

---

### Official Review · Reviewer_9FMk · 2026-03-12

**Soundness:** 2
**Presentation:** 3
**Significance:** 2
**Originality:** 2
**Overall Recommendation:** 4
**Confidence:** 4

**Summary:**

This work investigates using pre-trained large vision-language models (VLMs) to assist vision-language-action models (VLAs) in diagnosing and recovering from robot manipulation failures. The authors propose a new benchmark dataset that spans perception, planning, and
control failures, and provides annotations for task stages, fault types, and spatiotemporal repair strategies, and use this dataset to benchmark 20 existing VLMs on robot failure scenarios. This paper then proposes a VLM–VLA collaboration mechanism that achieves closed-loop diagnosis and recover; empirical results show the proposed mechanism improves manipulation task performance across both simulated and real world tasks.

**Compliance With Llm Reviewing Policy:**

Affirmed.

**Key Questions For Authors:**

1. What does latency look like during real world evaluations? To have a dynamic setting where an VLM need to track and correct failure behaviors during robot execution, it's unclear whether/how the hardware is paused during VLM generation.

2. What exactly is a 'conservative' model? There's a few mentions of an VLM being conservative (such as the Qwen3-235B model), is this due to the way a model was pre-trained, or just an after-the-fact description of the model's behavior when evaluated on the proposed tasks?

**Limitations:**

yes

**Strengths And Weaknesses:**

Strengths:
1. Good presentation quality that explains the fairly complex system well. The figures and plots are informative and help clarifying details of the overall proposed system workflow.

2. Extensive experimental results across different pre-trained models and various robot manipulation tasks.

3. The problem setting is valid and important for large-scale real robot system deployment. The VLM–VLA collaboration schema is practical and connects well with in-the-wild robot applications.

Weaknesses:
1. Although extensive, the empirical evaluation results between different VLM models are a bit inconclusive, and there is not a lot of insights into the nuanced differences between different types of model backbones and discussions on why might be the cause of performance differences.

2. System-level implementation details, especially for the real world experiments, are not fully documented and it's a bit unclear on the exact usability of the proposed sim and eval framework.

---

> ### Author Rebuttal · Authors · 2026-03-31
>
> W1:We believe this two-sided phenomenon reinforces our main claim: strong performance on current VLM benchmarks (e.g., VQA) does not necessarily translate into improved embodied manipulation. Through failure diagnosis and backtracking, we expose the gap between capabilities that are benchmarked and those that are actually useful for robotic repair. In real-robot experiments, VLM-based diagnosis brings both correction gains and mis-intervention costs. Unstable diagnosis can misclassify successful trajectories as failures, triggering unnecessary rollback and turning correct executions into failures. Yet in other cases, VLMs correctly detect local optima or erroneous execution paths, trigger backtracking, and convert failed trials into successful ones.
> Our results further suggest a capability decomposition in the VLM–VLA repair pipeline: subtask understanding, fault diagnosis, severity assessment, temporal localization, and spatial correction. Reasoning-oriented backbones (e.g., GPT/Gemini) perform better on fault understanding, severity judgment, and subtask progress reasoning. Qwen-style models show relative advantages in spatial-object errors and local geometric grounding. Temporal localization remains challenging for all models, though Gemini and Llama are relatively stronger. Most importantly, all general-purpose VLMs still struggle with physically calibrated correction: they can often explain faults better than they can generate stable, executable repairs. Dynamic correction also shows clear axis-wise bias (X/Y/Z), indicating that repair is not a single capability, but a composition of temporal grounding, spatial localization, and correction calibration.
> W2/Q1：We thank the reviewer for raising the latency issue. We agree that if a VLM is only applied to offline videos, the meaning of a “dynamic” setting would be limited. Our real-robot evaluation, however, explicitly incorporates VLM latency into a closed-loop pause-and-inspect protocol.
> Specifically, diagnosis is triggered only 3–6 times per task, rather than at every control step. In each cycle, the robot first executes under the VLA for a fixed 1-second window (20 FPS segment), after which both robot motion and VLA inference are paused. The collected video is then sent to the VLM for diagnosis and recovery decisions. If a recoverable failure is detected, rollback/correction is executed before the next cycle. Thus, VLM latency is part of the actual control loop rather than an ignored offline factor:
> $$
> T_{\text{loop}} = T_{\text{exec}} + T_{\text{vlm}} + T_{\text{rollback/correction}},
> $$
> where $T_{\text{exec}} = 1$ second, $T_{\text{vlm}}$ denotes the average inference latency of the model, and $T_{\text{rollback/correction}}$ represents the time required for the robot to execute rollback and corrective actions.
> This design is not only a practical way to handle latency, but also matches the problem structure. In our experiments, once the VLA enters an erroneous trajectory or local-optimum failure state, it often cannot recover by itself and may oscillate around the failure for longer than the cost of a few diagnostic pauses. Sparse VLM diagnosis therefore adds some waiting time, but can recover trajectories that would otherwise remain failed.
> We will add mean VLM latency measurements, more complete system details in the appendix, and release FaultEval to improve reproducibility.
> Q2：Conservativeness is defined in terms of operational behavior. In real-robot experiments, Qwen-235B achieves the highest precision (0.7273), but relatively low recall (0.2162) and a low FPR (0.2516). This indicates that once the model raises an alarm, it is usually reliable. However, it may miss certain failures. The FPR reflects the probability of misclassification that interrupts normal execution. A high FPR is particularly dangerous, because once a false alarm is triggered, the system may initiate stop / rollback / correction, thereby interrupting an originally correct execution.\\
> | Model | gemini-2.5-pro | gemini-2.5-flash | Llama-3.2-90B-Vision | GPT-5-2 | Qwen-2.5-VL-7B | Qwen3-VL-235B | Doubao-1-5-vision-pro | Doubao-Seed-1.6 |
> |---|---|---|---|---|---|---|---|---|
> | Time | 4.2s ± 0.7s | 2.1s ± 0.5s | 2.3s ± 0.6s | 4.7s ± 0.6s | 1.5s ± 0.4s | 2.9s ± 0.7s | 2.1s ± 0.2s | 2.0s ± 0.3s |

---

> > ### Author Rebuttal · Reviewer_9FMk · 2026-04-03
> >
> > Thank you to the authors for the additional detailed follow-ups. I will maintain the current recommendation.

---

### Official Review · Reviewer_shPv · 2026-03-13

**Soundness:** 2
**Presentation:** 3
**Significance:** 3
**Originality:** 3
**Overall Recommendation:** 3
**Confidence:** 4

**Summary:**

This paper asks whether VLMs can help diagnose and recover from manipulation failures in VLA systems. The main contribution is VLA-FixBench, a benchmark with 6,034 annotated failures spanning perception, planning, and control, together with multi-dimensional annotations over task stage, severity, temporal onset, and spatial correction. The paper also introduces FaultEval, a static-to-dynamic-to-real evaluation protocol, and a VLM-VLA closed-loop mechanism that predicts when to stop, where to roll back, and how to apply a coarse spatial correction.

**Compliance With Llm Reviewing Policy:**

Affirmed.

**Final Justification:**

Since the authors did not provide a response, I decide to maintain my original score.

**Key Questions For Authors:**

1.What is the intended takeaway from the recovery experiments? Right now I read them as evidence that the benchmark is useful, but not yet evidence that the proposed VLM-VLA loop is practically effective.

2.Can the authors decompose closed-loop failures more explicitly? I would especially like to know how much is due to temporal localization, how much to spatial correction, and how much to downstream execution mismatch.

3.The current repair interface is very coarse. Did the authors try any richer correction parameterization, or do they believe the current bottleneck is mostly in diagnosis rather than in the repair action space?

4.The paper emphasizes the human upper bound. Can the authors make the distinction between autonomous performance and human-assisted potential more explicit throughout the paper?

5.Since the benchmark is a central contribution, can the authors provide stronger evidence on annotation consistency for failure type, severity, and temporal/spatial labels?

**Limitations:**

yes

**Strengths And Weaknesses:**

I think the benchmark/evaluation side is the strongest part of the paper by a clear margin. The paper is asking a good question, and the proposed benchmark is richer than the usual success/failure view of embodied evaluation. The hierarchical failure taxonomy is sensible, and the temporal/spatial/semantic annotation design makes the benchmark more diagnostically useful than most existing VLA evaluations. I also like the static-versus-dynamic framing: the paper makes it clear that “being good at describing faults” and “being useful in a repair loop” are not the same thing. That is a worthwhile message for the community.

Where I become much less convinced is the recovery story. The paper’s framing suggests a meaningful step toward autonomous VLM-assisted repair, but the real-robot results are much more sobering. In the current review draft, the key issue was already visible: VLM-guided judges underperform the VLA-only baseline in real-robot success rate, and some models achieve high recall only by becoming unstable intervention agents. That makes me view the proposed loop less as a demonstrated recovery method and more as an experimental interface for studying where such recovery breaks down.

I also found the current method interface too coarse relative to the claim. The dynamic tuple is just stop time, rollback time, and a discretized 3D correction vector. When this fails, it is very hard to tell whether the problem is bad VLM reasoning, weak temporal grounding, poor spatial grounding, controller mismatch, or simply the limited action space of the repair mechanism itself. In other words, the benchmark is disentangling failures better than the repair system is. That mismatch is the main reason I land at weak reject rather than weak accept.

So my overall take is: strong benchmark paper, interesting diagnosis paper, but not yet a convincing autonomous recovery paper. If the title and framing were a bit more benchmark-centered and a bit less solution-centered, I would likely feel more positive about it. The paper is well written and easy to follow; my issue is with how the evidence is being interpreted, not with readability.

---

### Official Review · Reviewer_YiLZ · 2026-03-18

**Soundness:** 3
**Presentation:** 3
**Significance:** 3
**Originality:** 3
**Overall Recommendation:** 4
**Confidence:** 4

**Summary:**

The paper discusses the effectiveness of VLMs in assisting VLAs and addresses the important gap of benchmarking for VLMs for detecting failures for VLA execution. The authors introduce a VLA-FixBench, which is a fault evaluation dataset that covers perception, planning, and control failures, and also propose FaultEval, which is an evaluation framework that contains both static and dynamic failure scenarios.

**Compliance With Llm Reviewing Policy:**

Affirmed.

**Key Questions For Authors:**

1. There is a fixed list of failure modes listed. Are these failure modes coming from pre-existing human judgment, or from a human summary of the tasks and the robot execution traces used in the work's experiment? How generalizable are these failure modes to other tasks, skills, and environments outside the paper's domain?
2. The paper mainly does experiments on a limited set of domains, like LIBERO in simulation and 2 real robot experiments. Since VLMs are general-purpose, how would their performance be on tasks, e.g., in the Open-X dataset (https://robotics-transformer-x.github.io/), that are much more general?
3. What is the latency (inference time) for these VLM models?
4. How sensitive are the performance of VLMs with respect to prompt engineering?
5. For Figure 4. Faults correlation analysis: why is it that the graph is not symmetric? Does it mean that the failure occuring order matters for the two failure modes?

**Limitations:**

There is a missing limitation section for the paper. Main limitations are:
- no closed-loop evaluation of VLM as failure detectors in real-time execution
- limited environments and benchmarks
- limited tasks variety

**Strengths And Weaknesses:**

Strengths:
- The problem setup is technically sound and serves an important problem. Having VLMs perform failure detections is an important and timely topic.
- The presentation is clear with helpful visuals and good writing.
- Extensive experiments are conducted on a variety of VLMs backbones.

Weaknesses:
- Although there are dynamic benchmark environments used, it is more like static videos rather than having the failure detectors in the closed loop of agent execution. Therefore, problems like inference latency delay are not considered, which could be a serious problem in real-time deployment.
- The environments used are quite simple: only LIBERO is used in simulation, which is quite similar in the visual appearance for all LIBERO tasks, and only two real robot environments. It is unclear whether the same conclusions hold for more tasks.
- The tasks used consist of quite distinct, semantically discrete skills like grasping. For more complex and continuous tasks, like stirring, folding, etc., the tasks do not come with a clear semantics, and it is unclear how the VLMs act on these.

---

### Decision · Program_Chairs · 2026-04-30

**Decision:**

Accept (regular)

**Comment:**

This paper introduces VLA-FixBench, a 6,034-trajectory step-level benchmark with rich annotations of VLA failures (fault type, severity, subtask success, temporal onset, spatial correction), along with FaultEval, a three-stage evaluation protocol spanning static, simulated, and real-robot settings across 20 VLMs. The main contribution is the evaluation protocol itself: it reveals a systematic gap between strong VLM performance on VQA-style static diagnostics and weak embodied recovery capability, an important and underexplored negative result.

Reviewer scores were split but mildly positive (4/3/4/4, avg. confidence 4.25). Reviewer shPv described the benchmark as “the strongest part of the paper by a clear margin,” while questioning the autonomous-recovery framing. Reviewer mVfz maintained a weak accept despite raising substantive concerns about the reversible-MDP assumption and overly complex metric notation.

The authors noted that part of their rebuttal was submitted as AC-confidential comments and did not reach reviewers. The AC apologize for not being able to fix this issue earlier when it was initially raise. These unseen responses appear substantive, including additional experiments, clarifications, and candid discussion of limitations. Incorporating this context, the main remaining issue is framing: the paper overstates autonomous recovery, as headline gains reflect human-expert upper bounds rather than VLM-only performance. The work is better positioned as a diagnostic benchmark exposing the static–dynamic gap. A revision with clearer framing and more transparent evaluation would significantly strengthen the paper.